# Intensive Follow-Up Program and Oncological Outcomes of Biliary Tract Cancer Patients after Curative-Intent Surgery: A Twenty-Year Experience in a Single Tertiary Medical Center

Alessandro Rizzo [1], Riccardo Carloni [2,3], Giorgio Frega [4], Andrea Palloni [2,3], Alessandro Di Federico [2,3], Angela Dalia Ricci [5,*], Raffaele De Luca [6], Simona Tavolari [2] and Giovanni Brandi [2,3]

1. Struttura Semplice Dipartimentale di Oncologia Medica per la Presa in Carico Globale del Paziente Oncologico "Don Tonino Bello", IRCCS Istituto Tumori "Giovanni Paolo II", Viale Orazio Flacco 65, 70124 Bari, Italy; a.rizzo@oncologico.bari.it
2. Department of Specialized, Experimental and Diagnostic Medicine, University of Bologna, Via Giuseppe Massarenti, 9, 40138 Bologna, Italy; riccardo.carloni2@studio.unibo.it (R.C.); andrea.palloni@aosp.bo.it (A.P.); alessandr.difederico@studio.unibo.it (A.D.F.); simona.tavolari@unibo.it (S.T.); giovanni.brandi@unibo.it (G.B.)
3. Division of Medical Oncology, IRCCS Azienda Ospedaliero-Universitaria di Bologna, Via Albertoni, 15, 40138 Bologna, Italy
4. Osteoncology, Bone and Soft Tissue Sarcomas, and Innovative Therapies, IRCCS Istituto Ortopedico Rizzoli, 40136 Bologna, Italy; giorgio.frega2@unibo.it
5. Medical Oncology Unit, National Institute of Gastroenterology, "Saverio de Bellis" Research Hospital, Castellana Grotte, 70013 Bari, Italy
6. Department of Surgical Oncology, IRCCS Istituto Tumori 'Giovanni Paolo II', Viale Orazio Flacco, 65, 70124 Bari, Italy; r.deluca@oncologico.bari.it
* Correspondence: dalia.ricci@unibo.it

**Abstract:** Aim. The aim of this research was to assess the impact of an intensive follow-up program on BTC patients who had received surgery with curative intent at a tertiary referral hospital. Methods. BTC patients were followed-up every three months during the first two years after their first surgery and every six months from the third to the fifth post-operative year. Results. A total of 278 BTC patients who received R0/R1 surgery were included. A total of 17.7% of patients underwent a second surgery following disease relapse, and none of these patients experienced additional disease relapse. Conclusions. An intensive follow-up after surgical resection may help in the early identification of disease relapse, leading to early treatment and prolonged survival in selected cases.

**Keywords:** biliary tract cancer; cholangiocarcinoma; follow-up; recurrence; surveillance

## 1. Introduction

Biliary tract cancer (BTC) includes a group of heterogeneous and rare tumors with poor prognosis, encompassing gallbladder cancer (GBC), intrahepatic cholangiocarcinoma (iCCA), and extrahepatic cholangiocarcinoma (eCCA), with the latter further subdivided into perihilar (pCCA) and distal (dCCA) variants [1,2]. Overall, these malignancies account for approximately 10–15% of primary liver cancers [3]; unfortunately, potentially curative surgical resection is possible in only around 25% of BTC patients at diagnosis, and even following radical surgery, relapse rates remain high [4,5]. The majority of patients are diagnosed with locally advanced, unresectable, or metastatic disease, and palliative chemotherapy represents the standard of care in this setting [6].

Since the incidence of BTC has risen in the past few decades due to an increase in iCCA incidence, tertiary medical centers more than doubled their annual caseload during the period from 2000 to 2020. Consequently, we have witnessed an increased demand for post-treatment follow-up in this patient population. Based on these premises, follow-up is important for BTC since some patients have a high risk of cancer recurrence, and these recurrences may respond better to treatment if detected early [7,8]. However, no standardized surveillance strategies

have been defined with respect to the type or timing of visits, and this inconsistency is also evident in the wide range of existing practices [9]. In this study, we evaluated the impact of an intensive follow-up program in a large cohort of BTC patients who had received surgical resection with curative intent at a tertiary referral hospital.

## 2. Materials and Methods

### 2.1. Patients

The Area Vasta Emilia Nord Ethics committee (l184/2020) approved this study. An observational cohort study was conducted, and the medical records of all consecutive BTC (iCCA, eCCA, and GBC) patients treated with surgical resection at Policlinico Sant'Orsola Malpighi Hospital, Bologna, Italy, from January 2001 to October 2021 were retrospectively reviewed. All clinical and pathological information was sought, including demographic variables, underlying co-morbidities, surgical modality, laboratory information, pathological reports, pre- and post-operative therapies, and follow-up information.

### 2.2. Inclusion Criteria

- More than 18 years of age;
- Surgical resection for primary BTC (iCCA, eCCA, and GBC);
- Pathologically verified R0 or R1 resection.

### 2.3. Exclusion Criteria

- R2 resection;
- Known metastases;
- Synchronous cancer;
- Concurrent participation in other studies that affect the frequency and content of the follow-up program.

### 2.4. The Follow-Up Program

Each patient had fourteen follow-up visits over 5 years, as follows: BTC patients were followed-up every 3 months during the first 2 years after their operations and every 6 months from the third to the fifth post-operative year. At each follow-up visit, the patient was examined; blood work was obtained, including CEA and CA 19.9, and an abdominal/chest CT scan with IV contrast was done.

### 2.5. Statistical Analysis

The Kaplan–Meier method determined the BTC-specific overall survival (OS) and disease-free survival (DFS) status. $p$ values were two-sided, and $p$ values $\leq 0.05$ were considered statistically significant. The IBM®SPSS® Statistics software (release 22.0) was used to perform all statistical analyses.

## 3. Results

A total of 398 consecutive BTC patients received surgical resection; 40 patients were excluded due to insufficient data, leaving 358 patients. Macroscopic residual tumor was observed in 29 patients (who received first-line treatment, as reported in Figure 1); thus, 329 BTC patients underwent R0 ($n = 222$; 67%) or R1 ($n = 107$; 33%) surgery. Among these R0/R1 patients, 278 subjects started a follow-up program at the institution, while 51 patients were excluded due to various reasons (e.g., they were followed by another institution, etc.); thus, it was not possible to analyze the clinical outcomes of this patient population. The baseline characteristics of patients are reported in Table 1; 80% of this group (222/278) started the follow-up after adjuvant chemotherapy or chemoradiotherapy (Figure 1). The median age was 63 years (range 37–85 years), and 170 (61.1%) of patients were females. Overall, 116 (41.7%) and 126 (45.4%) patients had iCCA and eCCA, respectively (78 pCCAs and 48 dCCAs).

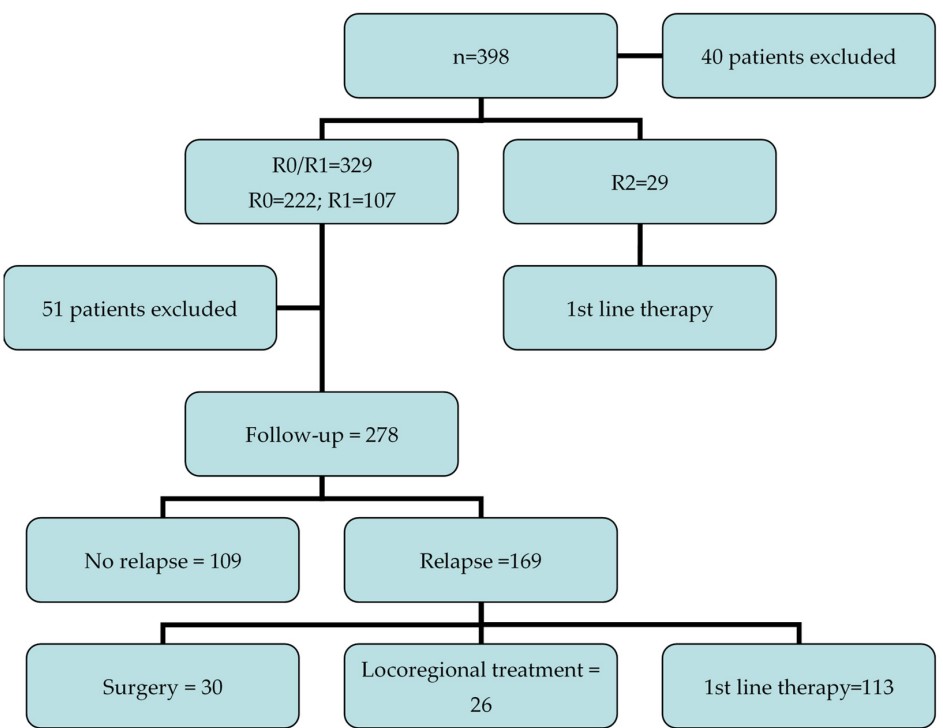

**Figure 1.** The study flow chart of the included patients.

**Table 1.** Baseline characteristics of patients.

| Patients | (*n* = 278) |
|---|---|
| **Sex** | |
| Male | 108 (38.9%) |
| Female | 170 (61.1%) |
| **Median age** | |
| | 63 years, range 37–85 |
| **Primary tumor site** | |
| iCCA | 116 (41.7%) |
| eCCA | 126 (45.4%) |
| GBC | 36 (12.9%) |
| **Grading** | |
| G1 | 24 (8.6%) |
| G2 | 136 (48.9%) |
| G3 | 102 (36.8%) |
| Not available | 16 (5.7%) |
| **Vascular infiltration** | |
| Yes | 96 (24.8%) |
| No | 39 (14.0%) |
| Not available | 143 (51.4%) |

Abbreviations: eCCA: extrahepatic cholangiocarcinoma; eCCA: extrahepatic cholangiocarcinoma; GBC: gallbladder cancer.

*Overall Survival and Disease-Free Survival*

At a median follow-up of 37.4 months, the median OS was 48.6 months in the overall population and 50.8 and 35.0 months in R0 and R1 patients, respectively. This difference

was statistically significant (*p* = 0.04) (Figure 2). Median DFS was 16.4 and 12.3 months in R0 and R1 patients, respectively (*p* = 0.02) (Figure 2). No relapse was observed in 109 out of 278 BTC patients (39%), while 61% of patients (169/278) experienced disease relapse; the liver was the most frequent site of relapse (78%), followed by the peritoneum and locoregional lymph nodes.

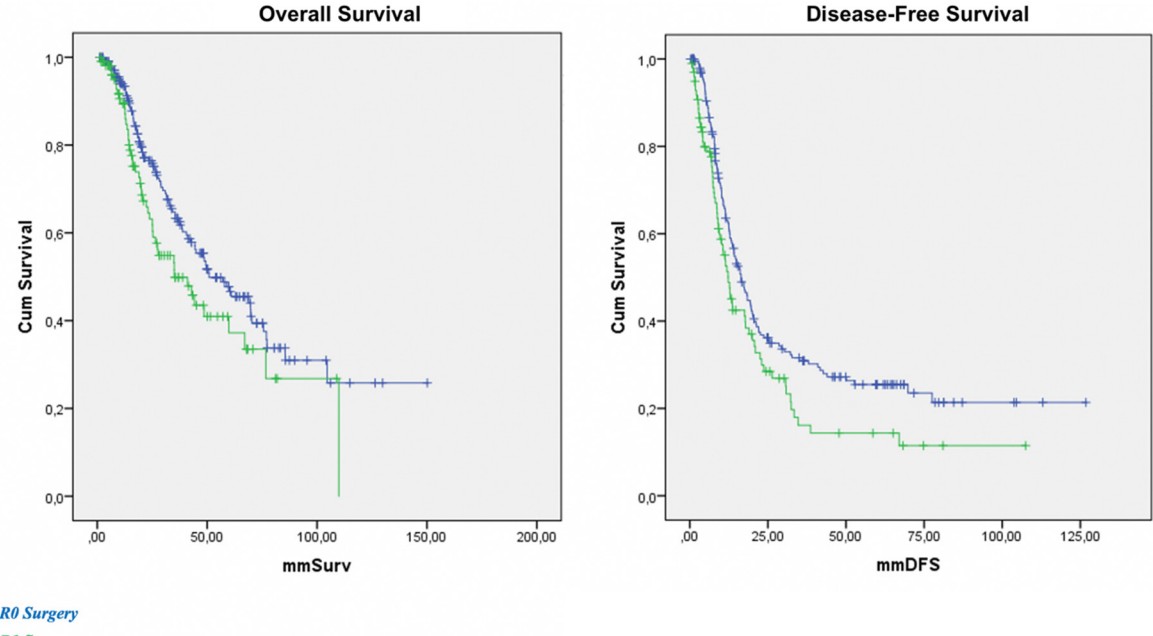

**Figure 2.** Overall survival (OS) and disease-free survival (DFS) in R0 (blue) and R1 (green) patients.

In terms of treatment at relapse, 17.7% (30/169) of patients underwent a second surgery (R0 = 14, R1 = 6, R2 = 3, exploratory laparotomy = 7) while 26 received a locoregional approach; the most common treatment was radiofrequency ablation (*n* = 22). None of the 14 BTC patients who received a second surgery that resulted in an R0 resection went on to experience disease relapse, and all these patients received the same follow-up program. Overall, 113 BTC patients received first-line chemotherapy following relapse, with gemcitabine–cisplatin as the most common first-line regimen (52%).

## 4. Discussion

The outcomes for BTC patients are poor, especially for patients at an advanced stage of disease, with a five-year overall survival rate of less than 20% [10,11]. Although these hepatobiliary tumors have been historically considered rare malignancies in Western countries, the incidence and mortality rate of BTC have risen in the past three decades, mainly due to the increased incidence of iCCA [12,13]. When feasible, early diagnosis and treatment remain the best therapeutic approach for BTC, and radical surgery remains the mainstay of curative treatment [14,15]. However, BTCs are frequently asymptomatic in the early stages and are diagnosed with locally advanced/unresectable or metastatic disease [16,17]. In addition, a large percentage (15–45%) of BTC patients presenting with apparently resectable disease are found to be unresectable during an exploratory laparotomy, and even following more aggressive surgical approaches, recurrence rates remain high [18]. In fact, although early-stage BTC may be effectively treated by surgery or ablative modalities, local or distant recurrence is frequent [19].

For most patients with cancer, there is a standard surveillance strategy to detect local recurrence or metastatic disease, as well as to allow appropriate treatment. These strategies are generally based on the tumor size, stage, and other clinicopathological features; however, no consensus currently exists for patients with BTC after surgery or locoregional therapy. In addition, there is a paucity of studies in the literature regarding

follow-up for patients with BTC; several authors have based their clinical choices on symptom inquiry, physical examination, serum CEA and CA19-9, and abdominal/chest computed tomography scanning [20,21]. The results of our 20-year single-center experiment involving 278 BTC patients, which is the first in the literature to be specifically focused on this topic, suggest that an intensive follow-up after surgical resection with curative intent could help in the identification of disease relapse, leading to early treatment and prolonged survival in selected cases, as reported in our BTC cohort treated with second surgical resection with negative margins.

BTCs are aggressive malignancies with high tumor recurrence, including in patients receiving radical surgery and adjuvant treatment. The results of our retrospective study highlight that an intensive surveillance program following surgical resection with curative intent could help in the identification of disease relapse in BTC. As regards the overall number of recurrent cases with curative-intent treatment, the follow-up program allowed us to detect a high number of relapses and more potentially curable diseases. At a median follow-up of 37.4 months, more than 60% of BTC patients had experienced disease recurrence. Therefore, it appears fundamental to design a close follow-up plan and strictly implement it in order to identify those patients with early symptoms and signs of recurrence and to use and apply appropriate treatments.

Our results further confirm the role of radical surgery, with R0 surgical resection representing the main prognostic factor in BTC patients. In fact, the median OS was 50.8 months in R0 BTC patients, compared with 35.0 months in R1 patients; similarly, R0 patients reported a median DFS of 16.4 months versus 12.3 months in R1 BTC patients. These findings support the role of an intensive follow-up program for BTC patients, with the aim of anticipating the diagnosis of recurrence and improving survival. In the patient population included in our study, 17.7% (30/169) of BTC patients underwent a second surgery (R0 = 14, R1 = 6, R2 = 3, exploratory laparotomy = 7); interestingly, no recurrences were observed in the 14 BTC patients treated with R0 surgery. In recent years, a growing number of studies have evaluated the role of repeated resections in BTC, especially in iCCA patients, with some of these reports suggesting the potential survival benefit of this surgical approach [22,23]. Our findings are consistent with these studies, highlighting that repeated surgical resection can be performed in selected BTC patients. While iterative resections represent well-established procedures for malignancies such as hepatocellular carcinoma and colorectal cancer liver metastasis, their role in BTCs remains to be clarified, and further investigations are needed [24–26].

Our study has its strengths and limitations. Among the former, our study is a single-institution experiment conducted in a highly respected academic center that serves as a referral/hub center for the management of BTC. Secondly, we included a relatively large number of BTC patients. Thirdly, all the included subjects were followed in an ordinary clinical setting; thus, the risk of selection bias was relatively low. The limitations include the retrospective nature of the study, the limited follow-up period (median of 37.4 months, which we acknowledge may be rather short), the lack of data on imaging such as magnetic resonance imaging (MRI), and the dependence upon the accuracy of documentation. In particular, the authors had access only to imaging and patient data stored at the hospital; since we included BTC patients from a large time period, this element could have introduced some bias. Thus, our findings should be interpreted with caution and should be considered hypothesis-generating only.

Several authors have recently predicted a marked increase in access to medical visits for BTC patients since the recognized incidence of iCCA continues to rise in a number of countries [27,28]. Therefore, it is necessary to evaluate the cost-effectiveness of follow-up programs, and the impact of resource- and time-consuming procedures, such as clinical visits and radiologic examinations, should be assessed. It is readily apparent that surveillance imaging is associated with repeated radiation exposure and may even lead to unnecessary biopsies or further, and even more resource-consuming, imaging. In our view, prospective

trials evaluating the most effective follow-up strategy for BTC are needed, both from the perspective of the impact on health care systems and from the clinical point of view.

## 5. Conclusions

Defining guidelines for follow-up in BTC patients receiving radical surgery is important from both economic and medical perspectives. In our single-institution experiment, an intensive clinical and radiologic follow-up allowed the identification of a high number of recurrences. Prospective trials are needed to define the most effective follow-up strategy for BTC patients, as well as the role of imaging, such as MRI and CT, in this setting.

**Author Contributions:** Conceptualization, A.R., R.C., G.F., A.P. and G.B.; methodology, A.R., R.C., G.F. and A.P.; software, A.R., G.F. and A.P.; validation, R.C., S.T., A.D.F., A.D.R. and R.D.L.; formal analysis, A.R. and G.B.; investigation, A.R., R.C., G.F. and A.P.; resources, all authors; data curation, A.R., G.F. and A.P.; writing—original draft preparation, A.R. and A.D.R.; writing—review and editing, all authors; visualization, all authors; supervision, R.D.L. and G.B.; project administration, A.R.; funding acquisition, none. All authors have read and agreed to the published version of the manuscript.

**Funding:** This research received no external funding.

**Institutional Review Board Statement:** The study was conducted in accordance with the Declaration of Helsinki and approved by the Area Vasta Emilia Nord Ethics committee (l184/2020).

**Informed Consent Statement:** Informed consent was obtained from all subjects involved in the study.

**Data Availability Statement:** The data presented in this study are available on request from the corresponding author.

**Conflicts of Interest:** The authors declare no conflict of interest.

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
