# Peer review of "Intensive Follow-Up Program and Oncological Outcomes of Biliary Tract Cancer Patients after Curative-Intent Surgery: A Twenty-Year Experience in a Single Tertiary Medical Center"

_curroncol, doi:10.3390/curroncol29070402_

Round 1
Reviewer 1 Report
The paper is well written and it's clear, certainly this is a single center review of their clinical cases but it has not particular novelty. The authors remark a well know truth about BP cancers, hoverer I recommend it for pubblicato with average priority
Author Response
Dear Reviewer,
Thank you so much for the time spent revising our work and thank you for appreciating our efforts.
Best regards
Reviewer 2 Report
The manuscript presents the results of a cohort study, that evaluates the impact of an intensive follow-up program in patients who underwent surgical resection with curative intent for biliary tract cancer.
The authors show that an intensive surveillance program could help in the identification of disease relapse.
The article is well written and well structured. The topic is interesting. No standardized surveillance strategy has been yet defined with respect to type and timing of visits for biliary tract cancer patients. It is an important research area.
As the authors underline, their study is a retrospective single-institutional experience, but they focus on an important research area. Developing a specific surveillance strategy to detect local recurrence or metastatic disease for patients undergoing surgical resection for biliary tract cancer is important to allow appropriate treatment. There is little evidence in medical literature and no consensus about the best surveillance strategy for these patients.
If available, it would be interesting to have more information about characteristics of the tumors, such as size and presence of vascular infiltration, and type of surgery (e.g. cholecystectomy, liver resection, pancreatoduodenectomy).
In my opinion, the article can be published.
Author Response
Dear Reviewer,
Thank you so much for the time spent revising our work and thank you for your valuable comments.
As suggested, we added more information to Table 1, by including several features (e.g., vascular infiltration, grading, etc.). All our changes have been reported in red color in the revised manuscript.
Best regards
Reviewer 3 Report
The manuscript describes different tumour entities that should be analyzed seperately. The median follow-up is 37.4 mo that is rather short. A study flow chart would help to identify included patients and their according treatments, drop-outs, recurrence rates and treatment of recurrence.
Author Response
Dear Reviewer,
Thank you so much for the time spent revising our work and thank you for appreciating our efforts. We modified the study flow chart, as suggested (blue).
We hope the revised paper will better suit the journal.
Best regards
Reviewer 4 Report
I feel your paper is interesting, especially for demonstrating that recurrences are opportunities to achieve cure and not, as commonly considered, indications of failure.
I have a number of editorial suggestions which have more to do with getting the English (American) understandable. I may take liberties as I may misunderstand what was meant.
I have been using MRI exams more frequently now that had been the case 5 years ago, but you do not mentioned their use at all. We have found Diffusion Weighted Imaging very helpful in finding liver metastases not clearly seen on CT. I would like you to address that modalities use in the future in your program.
Editing comments:
Page 1, line 25: Please remove "the" prior to 17.7%.
Page 1, line 42: Consider rewriting the first sentence to read, "in particular due to an increase in the incidence of iCCA, tertiary...".
Page 1, line 45: Consider rewriting the third sentence to read, "follow-up is important for BTC since these patients are at high risk...".
Page 2, line 54: Consider rewriting the first sentence to "The local Institutional Review Board approved this study."
Page 2, line 58: The third sentence of paragraph 2.1 is confusing to me and needs to be rewritten. My understanding of the intent of the sentence would lead me to rewrite the beginning to be, "All clinical and pathological information was sought, including..."
Page 2, line 75: I want you to be more specific about what you do at each follow-up visit. I suggesting rewriting the second sentence of paragraph 2.4, to be, "At each follow-up visit, the patient was examined; blood work was obtained including CEA and CA 19.9; and an abdominal/chest CT scan with IV contrast was done." This is what I assumed you to have meant.
Page 2, line 83: In this paragraph, you detailed the patients you studied, and pointed out that you had found 358 patients, but 40 patients were excluded due to missing variables, 29 were excluded due to leaving behind macroscopic residual tumor, and 51 patients were not involved in the follow-up program. I would like to know what happened to these 120 patients, which is a large percentage of the original patients you identified. In particular, what data were missing from the 40 patients that caused you not to include them. The other side of that question is how complete is the data on the patients you kept, i.e., did they all have 14 follow-up appointments, 14 sets of blood work, and 14 CT scans?
Another question is what was the fate of the 29 patients with macroscopic residual cancer? One would have expected that they all succumbed, but was that the case?
Finally, there were 51 patients who had sufficient data and had R0 or R1 resections but did not do the follow-up with your program. Why was that? You had done the surgery and so why did they not follow up with you? What happened to them? Did they have a similar survival curve as they other patients?
Page 2, line 83: I suggest rewriting the second half of the first sentence to "40 patients were excluded due to insufficient data, leaving 358 patients."
Page 2, line 85: I suggest replacing "while" with "thus".
Page 2, line 88: I suggest removing the "the" and starting the fifth sentence with "80%".
Page 3, line 104: I am not certain you need to write out the y-axis legend on either figure as the title of the figure is self-explanatory. The x-axis legend should be "Time (months)".
Page 3, line 106: I suggest removing the "the" before "relapse" and the "the" before "17.7%", and adding an "a" before "second".
Page 3, line 108: I suggest changing the comma after approach to a semicolon and rewrite the last part of the sentence to be "the most common treatment was radiofrequency ablation (n=22)."
Page 3, lines 108-109: I suggest rewriting the second sentence to be, "None of the 14 BTC patients who received a second surgery which resulted in a R0 resection went on to experience disease relapse."
Page 3, line 111: I suggest replacing "used" with "common".
Page 4, line 117: I suggest rewriting the end of the second sentence to be ", mainly due to the increased incidence of iCCA [12, 13]".
Page 4, line 121: I suggest changing "a proportion ranging from 15 to 45%" to "a large percentage (15-45%)".
Page 4, line 125: I suggest changing "is a frequent event" to "is frequent".
Page 4, line 126: I suggest rewriting the first two sentences to: "For most patients with cancer, there is a standard surveillance strategy to detect local recurrence or metastatic disease, as well as, to allow appropriate treatment. These strategies are generally based on the tumor size, stage, and other clinico-pathological features; however, no consensus currently exists for patients with BTC after surgery or locoregional therapy."
Page 4, line 140: I suggest replacing "also" with "even".
Page 4, line 156: I suggest removing the "the" prior to "17.7%".
Page 4, line 165: I suggest replacing the "are" with "is".
Page 5, line 169: I suggest replacing "an overall" with "a relatively".
Author Response
Dear Reviewer,
Thank you so much for the time spent revising our work and thank you for all your valuable comments and suggestions.
As you may see, we extensively modified the paper according to your suggestions. All our changes have been reported in green color. For example, we replaced several sentences and words, as suggested, and we better explained some crucial points, including the lack of data regarding MRI and its possible current and future role in this setting.
Thank you again. We hope the revised paper will better suit the journal.
Best regards